# An International Inter-Consortium Validation of Knowledge-Based Plan Prediction Modeling for Whole Breast Radiotherapy Treatment

**DOI:** 10.3390/cancers17213576

**Published:** 2025-11-05

**Authors:** Lorenzo Placidi, Peter Griffin, Roberta Castriconi, Alessia Tudda, Giovanna Benecchi, Mark Burns, Elisabetta Cagni, Cathy Markham, Valeria Landoni, Eugenia Moretti, Caterina Oliviero, Giulia Rambaldi Guidasci, Guenda Meffe, Tiziana Rancati, Alessandro Scaggion, Karen McGoldrick, Vanessa Panettieri, Claudio Fiorino

**Affiliations:** 1Department of Diagnostic Imaging, and Oncological Radiotherapy, Fondazione Policlinico Universitario Agostino Gemelli, IRCCS, 00195 Rome, Italy; lorenzo.placidi@policlinicogemelli.it; 2Alfred Health Radiation Oncology, The Alfred Hospital, Melbourne, VIC 3004, Australia; pete.griffin@genesiscare.com; 3GenesisCare, Cabrini Hospital, Melbourne, VIC 3144, Australia; 4Medical Physics Department, IRCCS San Raffaele Scientific Institute, 20132 Milano, Italy; tudda.alessia@hsr.it (A.T.); fiorino.claudio@hsr.it (C.F.); 5Medical Physics Department, University Hospital of Parma AOUP, 43126 Parma, Italy; gbenecchi@ao.pr.it; 6Department of Radiation Therapy Services, Peter MacCallum Cancer Centre, Melbourne, VIC 3000, Australia; mark.burns@petermac.org (M.B.); cathy.markham@petermac.org (C.M.); karen.mcgoldrick@petermac.org (K.M.); 7Medical Physics Unit, Department of Advanced Technology, Azienda USL-IRCCS di Reggio Emilia, 42122 Reggio Emilia, Italy; elisabetta.cagni@ausl.re.it; 8AO San Camillo Forlanini, UOC Fisica Sanitaria, 00152 Rome, Italy; vlandoni@scamilloforlanini.rm.it; 9Department of Medical Physics, University Hospital, 33100 Udine, Italy; eugenia.moretti@asuiud.sanita.fvg.it; 10University Hospital ‘‘Federico II”, 80131 Napoli, Italy; caterina.oliviero@gmail.com; 11UOC di Radioterapia Oncologica, Ospedale Isola Tiberina—Gemelli, 00186 Rome, Italy; giulia.rambaldiguidasci@fbf-isola.it; 12Operative Research Unit of Radiation Oncology, Fondazione Policlinico Universitario Campus Bio-Medico, 00128 Rome, Italy; g.meffe@policlinicocampus.it; 13Data Science Unit, Fondazione IRCCS Istituto Nazionale dei Tumori, 20133 Milano, Italy; tiziana.rancati@istitutotumori.mi.it; 14Medical Physics Department, Veneto Institute of Oncology IOV–IRCCS, 35128 Padua, Italy; alessandro.scaggion@iov.veneto.it; 15Department of Physical Sciences, Peter MacCallum Cancer Centre, Melbourne, VIC 3000, Australia; vanessa.panettieri@petermac.org; 16Department of Oncology, The University of Melbourne, Parkville, VIC 3010, Australia; 17School of Translational Medicine, Monash University, Melbourne, VIC 3800, Australia

**Keywords:** international consortia, knowledge-based plan prediction, models’ transferability, whole-breast radiotherapy

## Abstract

**Simple Summary:**

This study explores whether treatment planning models developed in one country can be effectively used in another, helping to improve and share expertise in cancer radiotherapy. The researchers focused on right whole breast radiation treatments, creating multiple models from a large national group and testing them on patients from a different international group. The analysis examined how accurately the models predicted radiation doses to key organs, such as the lungs, and whether these predictions aligned with actual clinical data. The findings showed that most models worked well across different patient groups, with good accuracy in predicting lung doses. This suggests that knowledge-based planning models can be reliably shared between institutions, potentially saving time, improving treatment quality, and reducing variability in patient care. Such model sharing could strengthen collaboration between centers and accelerate advancements in radiotherapy planning worldwide.

**Abstract:**

Background: Knowledge-based (KB) planning is a promising approach to model prior planning experience and optimize radiotherapy. To enable the sharing of models across institutions, their transferability must be evaluated. This study aimed to validate KB prediction models developed by a national consortium using data from another multi-institutional consortium in a different country. Methods: Ten right whole breast tangential field (RWB-TF) models were built within the national consortium. A cohort of 20 patients from the external consortium was used for testing. Transferability was defined when the ipsilateral (IPSI) lung first principal component (PC1) was within the 10th–90th percentile of the training set. Predicted dose–volume parameters were compared with clinical dose–volume histograms (cDVHs). Results: Planning target volume (PTV) coverage strategies were comparable between the two consortia, even though significant volume differences were observed for the PTV and contralateral breast (*p* = 0.002 and *p* = 0.02, respectively). For the IPSI lung, the standard deviation of predicted mean dose/V20 Gy was 1.13 Gy/2.9% in the external consortium versus 0.55 Gy/1.6% in the training consortium. Differences between the cDVH and the predicted IPSI lung mean dose and the volume receiving more than 20 Gy (V20 Gy) were <2 Gy and <5% in 88.7% and 92.3% of cases, respectively. PC1 values fell within the 10th–90th percentile for ≥90% of patients in 6/10 models and 65–85% for the remaining 4. Conclusions: This study demonstrates the feasibility of applying RWB-TF KB models beyond the consortium in which they were developed, supporting broader clinical implementation. This retrospective study was supported by AIRC (Associazione Italiana per la Ricerca sul Cancro) and registered on ClinicalTrials.gov (NCT06317948, 12 March 2024).

## 1. Introduction

Breast cancer remains the most common malignancy in women worldwide and a leading cause of cancer mortality [1]. Contemporary management combines surgery with adjuvant radiotherapy (RT) to reduce local recurrence and improve survival [2]. However, ‘manual’ tangential field (TF) planning is time-consuming and prone to inter-planner and inter-institution variability, with a potential downstream impact on normal tissue doses and plan quality. Automated approaches—including protocol-based, multicriteria, and knowledge-based (KB) planning—offer standardization and efficiency, but their reliability across countries and institutions remains a critical, under-studied question that our work addresses [3].

The standard treatment for breast cancer typically involves radiation therapy (RT) administered as adjuvant therapy following surgery, often combined with chemotherapy and/or hormonal therapy [4,5]. Currently, whole breast irradiation is a well-established therapeutic approach, with tangential fields (TF) being the predominant and widely utilized method, primarily using manual optimization. Manual TF planning and optimization introduces clinically relevant variability in target coverage and heart/lung doses, is highly dependent on individual expertise and time pressure, and creates throughput bottlenecks that limit standardization and can delay the start of the treatment. To avoid this highly labor-intensive manual process that also relies on the expertise of the planner, various automated planning approaches have been proposed, including protocol-based, multicriteria optimization, and knowledge-based (KB) optimization [6]. Among available automation paradigms, protocol-based/template methods promote consistency but are inherently site-specific and less adaptive to patient geometry. For example while multi-criteria optimization (MCO) can expose Pareto-optimal trade-offs it continues to heavily rely on user interaction, preserving inter-planner variability. In addition, deep-learning based planning solutions show promise but currently require large, harmonized training datasets and are less widely deployed across our centers worldwide. In contrast, KB optimization is a data-driven automated planning method that utilizes prior knowledge and experience to predict an “optimally” achievable dose for a new patient from a similar population. This approach offers several advantages, such as reducing or eliminating inter-planner variability [7], reducing planning time [8], minimizing suboptimal plans [9], avoiding dose distributions that deviate far from the previous clinical experience, permitting unbiased plan quality comparison [10,11,12], and aiding in training and education [13]. However, there are also cons, including the ‘garbage in, garbage out’ risk if the plans included in the training datasets are not of the highest quality, the time required for generation/validation to translate the KB method into effective and automatic plan solutions, and the need for updates in case of protocol modifications [14,15,16,17]. Several single institutions, various publications have demonstrated overall plan quality and efficiency improvements, ref. [7,18,19,20,21]. When applied on a multi-institutional scale, KB planning can achieve several notable outcomes. Ideally serve as a strong rationale for in silico plan comparison and QA, it is powerful for QA in clinical trials, it can assist in patient treatment modality selection (e.g., particle vs. photon), and allows the sharing of KB models, especially with the commercialization of KB tools. However, challenges still exist in large-scale KB model implementation [22], including inter-institutional protocol variability (dose fractionation, technique, etc.), variability in clinical target volume (CTV), planning target volume (PTV), and organs at risk (OARs) definition and contouring between institutes, the lack of a well-established strategy/workflow for KB methods, and the interchangeability of KB methods between different centers.

It is quite well-established that while plan quality may largely vary between planners of the same institutione, much larger variations can be expected among different clinical institutions [23,24,25,26,27,28]. A major limitation of KB approaches is that inter-institutional variability is not considered in the training phase, making the possibility of applying a KB model outside the generating center challenging. Recent publications (Appendix A) have demonstrated the feasibility of sharing models across institutiones. However, in order to share models, it is first necessary to assess the degree of plan transferability between different institutes; that is the primary aim of this study, which consists of an inter-consortia validation of DVH prediction models.

Multi-institutional KB plan prediction models were previously generated to assess differences in plan performance within a national consortium for right whole breast (RWB) irradiation using TF. For this reason the current investigation aimed to quantify the resulting KB plan predictions in a separate multi-institutional consortium in another country to assess their “geographical” transferability. Due to possible variations in patient anatomy from different countries and then PTV/OARs segmentation variability, assessing models’ transferability is crucial. To our knowledge, it is the first experience of KB model validation on whole breast irradiation outside of national boundaries.

## 2. Materials and Methods

### 2.1. KB Model Definition, Model Set Criteria, and Intra-Consortium Validation

This retrospective study was conducted under the MIKAPOCo (Multi-Institutional Knowledge-based Approach for Planning Optimization for the Community) project. The study was supported by a grant from AIRC (Associazione Italiana per la Ricerca sul Cancro) and approved by the Ethics Committee of the host institution (IRCCS Ospedale San Raffaele, protocol number IG23150). The project has been registered on ClinicalTrials.gov (trial registration number: NCT06317948; date of registration: 13 March 2024). The aim of MIKAPOCo was to build consistent KB model libraries and incorporate inter-institutional variability into plan prediction, focusing on the whole breast (WB) irradiation scenario. MIKAPOCo developed ten distinct KB models following standardized criteria, specifically for patients treated with RWB-TF using manually optimized wedges or field-in-field (FiF) techniques [28]. Each of the ten centers built their individual KB models using the Model Configuration tool of RapidPlan (RP, Varian Medical Systems, Inc., Palo Alto, CA, USA) following common criteria. RapidPlan configures a model using existing clinical plans, combining principal component analysis (PCA) and regression techniques. Using the modeled data, the tool generates a dose–volume histogram (DVH) prediction for a new patient case optimization. Different Eclipse software versions (from V13.5 to V16.1) were utilized based on the availability at each institute. The models were trained and validated using a common methodology, requiring a minimum number of patients (>70), adherence to national guidelines for contouring (CTV/PTV/OARs), consistency in techniques (wedged or FiF), outlier elimination criteria, and evaluation of the goodness of the regression, which were jointly discussed and defined, as reported in Tudda et al. [26] (also in Appendix A). As reported in Tudda et al. [28], each model was tested on a cohort of 20 patients, randomly selected from ten MIKAPOCo institutes (two per center), to evaluate the models’ variability and transferability. To enhance robustness, we intentionally replicated the methodology of Tudda et al. [28], treating their results as the starting point and proof of concept, thereby enabling international, inter-consortium validation of knowledge-based plan prediction modeling for whole breast radiotherapy.

### 2.2. Comparison of the Intra- and Inter-Consortium Models’ Prediction Variability and Transferability

An external model validation test was performed using 20 patients from another national consortium, the Victorian Public Sector RapidPlan Group (VPSRG). Within this consortium, two independent centers each contributed 10 patients for validation. The assessment included evaluating dose–volume model predictions and the transferability of the models between consortia. Each of the ten models from MIKAPOCo was tested on the 20 patients provided by VPSRG (Figure 1).

The VPSRG patients’ cohort was initially planned and treated with TF using manually optimized wedges or FiF techniques, using electronic tissue compensation (eComp) and skin flash. Plans’ clinical goals are reported in Appendix A. For each patient, 10 RP estimations were generated by setting energy and beam arrangements consistent with MIKAPOCo’s recommendations, based on a dedicated protocol (Appendix A). The predicted DVHs generated by each of the 10 MIKAPOCo models for all OARs, including heart, contralateral breast, ipsilateral (IPSI) lung, and contralateral lung, were compared, providing the first quantification of inter-consortia variability in plan performances. All models were run using the most commonly employed fractionation regimen of 40 Gy in 15 fractions. For each OAR, prediction bands were exported by running a dedicated script.

PTVs were compared between the two consortia (CTV–PTV margins used for both the consortia were 5 mm isotropically in all directions). The percentage of the VPSRG’s PTVs inside the MIKAPOCO’s models in terms of joint target volume was also analyzed (specific parameter provided by the RapidPlan software (V16.1), defined as the volume of all matched target structures in cm^3^), to quantitatively evaluate differences in contouring and dataset variations due to anatomical differences. Dose–volume parameters were analyzed using PTV metrics (V95%, V105%, D1%, and D99%) to assess major variations in the planning approach.

To determine if model predictions were influenced by overlap in geometry and anatomical features from the original training cohort, the validity of the fit for each patient’s geometrical features in the VPSRG test set was examined. Additionally, the estimated statistics for each selected OAR, particularly the ipsilateral lung, heart, contralateral breast, and lung, were analyzed using RapidPlan’s Model Configuration tools. For inter-consortia transferability, ipsilateral lung DVH predictions were categorized considering the V20%–V80% range as optimal (cDVH within the predicted DVH band), suboptimal (cDVH below the band), improved (cDVH above the band), and failed (predicted DVH band strongly disagrees with cDVH). Examples are provided in Appendix A.

The predicted ipsilateral lung mean dose and V20 Gy were then calculated to quantify the variability of plan predictions between consortia. Additionally, for the VPSRG test dataset, the clinical DVH (cDVH) was compared with the predicted DVH. For each patient (i = 1, …, 20) and each model (j = 1, …, 10), the mean predicted DVH values (described in Equation (1) as $\underline{{DVH}_{i}}$) were extracted as the mean values between the upper and lower predicted DVH bands, as described in Tudda et al. [26], and compared with the clinical DVH (cDVH). The corresponding inter-institute standard deviations (SD) were assessed as described in Equation (2).(1)$$\underline{{DVH}_{i}}=\frac{1}{10}\sum_{j=1}^{10}\;\;\;{DVH}_{ij}$$(2)$$\sigma_{i}=\sqrt{\frac{\sum_{j=1}^{10}\;\left({\ddot{DVH}}_{ij}-\underline{DVHi}\right)}{9}}$$

The overall average DVH (including inter-institute variability) over all 20 patients was estimated. Subsequently, the overall mean standard deviation values for each corresponding mean DVH (DVHint), representing inter-institute variability, were calculated as follows:(3)$${DVH}_{int}=\;\frac{1}{20}\sum_{i=1}^{20}\;\left(DVHi\right){DVH}_{int}=\;\frac{1}{20}\sum_{i=1}^{20}\;\;\;\underline{{DVH}_{i}}$$(4)$${SD}_{int}=\frac{1}{20}\sum_{i=1}^{20}\;\;\;\sigma_{i}$$

Therefore, Equations (1) and (2) describe the overall average and mean standard deviation DVH over the different models, while Equations (3) and (4) describe these values over the patient cohort. For interpretability, we defined Δ = cDVH − predicted. Positive Δ indicates that clinical dose/volume exceeds the prediction. We report the proportions within ± 5% (Δ V20 Gy) and ±2 Gy (Δ mean dose).

Once the target coverage approach and the predicted mean doses (and their standard deviations) were compared and found to be in agreement, model transferability was quantified. This was done by counting the number of cases in which the ipsilateral lung PC1 fell within the 10th and 90th percentiles of the training set. Inter-institute transferability was assessed using the methodology from Tudda et al. [28] and compared with results from an intra-validation cohort of 20 patients, randomly selected from ten MIKAPOCo institutes (two per center). In Tudda et al. [28], each institution’s model was tested on 18 out of 20 patients, excluding those from the model’s originating center. No formal statistical power calculation was performed because the primary endpoint—across both intra- and inter-consortium cohorts—was the estimation of model transferability/feasibility metrics (Δ distributions, PC1-band inclusion) across ten site-specific models, not at the patient-level hypothesis testing.

### 2.3. Terminology and Analysis Context

To aid clarity and reproducibility, we provide concise definitions of core terms—MIKAPOCo, VPSRG, model prediction, variability, and transferability—and specify the analytic perspective used to distinguish intra- versus inter-consortium contexts throughout this study.

MIKAPOCo (source consortium). Multi-institutional cohort from which the 10 site-specific KB models used in this study were trained/derived. Unless otherwise stated, “intra-consortium” refers to analyses where a MIKAPOCo-derived model is applied to MIKAPOCo patients (same consortium as model origin).

VPSRG (external consortium). Independent multi-center cohort used as the external application/validation set. In this manuscript, applying a MIKAPOCo-derived model to VPSRG patients is termed inter-consortium (different consortium from model origin).

Model prediction. The dose–volume prediction produced by a knowledge-based model for a given patient’s anatomy/beam geometry, reported as DVH quantities for OARs and targets. Importantly, this study evaluates prediction (not re-optimization or delivery).

Variability.
-Patient-level clinical variability: Spread of delivered clinical DVH metrics within a cohort (anatomy, geometry, and practice effects; summarized by mean value and SD).-Prediction residual variability: Dispersion of Δ = clinical − predicted; reported within accuracy bands (±2 Gy mean dose; ±5% V20 Gy) and tail checks.-Model-to-model variability: For each patient, spread of predictions across the 10 site-specific models, aggregated to quantify dependence on model provenance/training distribution.

Transferability. The extent to which a model maintains accuracy and consistency when applied outside of its training distribution. We assess intra-consortium transferability (model to same-consortium patients) and inter-consortium transferability (model to other-consortium patients). All “intra/inter” labels are defined relative to the model’s training origin (“source model to target cohort”).

## 3. Results

VPSRG’s and MIKAPOCo’s test dataset (20 test patients each) were compared in terms of PTV metrics, as depicted in Figure 2. With regard to the PTV, the clinical VPSRG dataset exhibited homogeneity in terms of V95%, D1%, V105%, and D99%, except for some outliers (due to clinical requirements), as shown in Figure 2.

PTVs and OARs’ volumes were also compared between the twenty VPSRG test datasets and the twenty MIKAPOCo test datasets (Figure 3). Volume differences have been evaluated using the Mann–Whitney U test (significant differences if *p* < 0.05). Similar volumes (mean volume ± SD) have been observed for the heart in the MIKAPOCo test dataset and the VPSRG test dataset (656.3 ± 348.1cc vs. 609.50 ± 93.53cc *p* = 0.5). This behavior was similar for the contralateral lung (MIKAPOCo = 1199.5 ± 244.9cc vs. VPSRG = 1179.06 ± 442.22, *p* = 0.1) and for the IPSI lung (MIKAPOCo = 1501.1 ± 262.9cc vs. VPSRG = 1449.0 ± 470.8cc, *p* = 0.1). On the other hand, PTVs and left breasts show statistically significant differences in terms of volume values: PTV MIKAPOCo = 677.5 ± 363.7cc vs. VPSRG = 1151.9 ± 605.9cc (*p* = 0.002) and left breast MIKAPOCo = 685.46 ± 543.4cc vs. VPSRG = 1043.9 ± 503.0cc (*p* = 0.02).

To better quantify possible differences in contouring and/or volumes, the percentage of the VPSRG’s PTVs inside the prediction of the MIKAPOCo’s models was on average 51%. If we consider each individual MIKAPOCo models, the minimum and maximum percentage of the VPSRG’s PTVs inside the prediction was 30% (model from Institute 8) and 70% (model from Institute 5), respectively.

The ipsilateral lung DVH predictions from MIKAPOCo’s KB models were categorized and quantified, yielding 76% optimal predictions, 15% improved predictions, 6% suboptimal predictions, and 3% of failed predictions (see Appendix A). Cases defined as failed predictions were then excluded from the remaining analysis.

Table 1 reports the predicted mean doses for each OAR (heart, contralateral breast, ipsilateral, and contralateral lung) along with their SD for both national consortia’s test datasets. Regarding the prediction of ipsilateral lung V20 Gy, SD_int_ was 2.9% for the VPSRG test datasets, compared to 1.6% for the MIKAPOCo test datasets.

Across the VPSRG test set and the ten MIKAPOCo models, differences (Δ = cDVH − predicted) in ipsilateral lung V20 Gy and mean dose are summarized, respectively, in Figure 4A,B. Overall, 92.3% of Δ V20 Gy values were within ± 5% and 88.7% of Δ mean dose values were within ± 2 Gy. We report 95% binomial confidence interval for these proportions for transparency (pooled count basis): 88.7% corresponds to approximately 83–93%, and 92.3% to approximately 88–95%. These intervals are illustrative and likely conservative given within-patient and within-model clustering.

In the intra-consortium validation (Figure 5-left), MIKAPOCo demonstrated high inter-institute interchangeability, with ipsilateral lung PC1 values within the 90th percentile in less than 10% of test patients for 9 out of 10 models. When considering the inter-consortium validation patients’ cohort from VPSRG (Figure 5-right), PC1 values for each MIKAPOCo model were within the 90th percentile in less than 10% of the test patients for 6 out of 10 models. Institution 6 exhibited poor transferability, both for intra-consortium validation within MIKAPOCo (7 out of 18 patients) and inter-consortium validation within VPSRG (7 out of 20 patients). Additionally, worse transferability was observed for Institutes 10 and 7 in the international validation done by the VPSRG (5 and 4 out of 20 patients, respectively). VPSRG shows a significantly higher PC1-Outside fraction than MIKAPOCo across models (Wilcoxon signed-rank test *p* = 0.0207).

## 4. Discussion

National KB face challenges in harmonizing plan libraries and manage inter-institution variability in treatment planning because model-building methods are currently not standardized. International comparisons could enable common modeling and cross-validation standards, enabling robust, population-wide assessments that account for contouring and anatomical differences.

The presented study constitute the first assessment of international consortia KB prediction models on WB irradiation, anticipating subsequent demonstrations of their quantified transferability. Beyond the known benefits of KB models for “in silico” plan comparisons [29,30], quality assurance, and clinical trials/audits [31,32,33,34,35,36], this assessment offers national consortia or single institutions a valuable benchmark to adopt automated planning without requiring extensive expertise, promoting broader use across diverse clinical settings. KB models developed by cross-institutional groups, leveraging large-scale data, have the potential to enhance the homogeneity of plan quality [37,38,39] and can be valuable in ensuring compliance with predefined planning criteria performances within the context of clinical trials [25,26,40]. Recently, a similar study by Jain et al. [41] evaluated and quantified the universal applicability of two cervical cancer KBP models developed in two different institutions (Asia and Europe) based on the EMBRACE-II protocol, using respective patients’ plans. These two KB models were exchanged between three institutions with different geo-ethnic populations and validated on reference manual plans, demonstrating cross-continental applicability of KBP for cervix cancer. In contrast, our study focuses on right whole breast tangential fields, evaluates ten models assembled within a national consortium, and assesses transferability of DVH predictions using a PC1-based within-band criterion rather than full plan re-optimization. Methodologically, Jain’s output involves plan quality from generated plans, whereas ours is prediction fidelity and model portability; accordingly, the comparative emphasis is different (plan performance vs. prediction agreement). Despite these differences in disease site, technique, model provenance, and evaluation endpoints, both studies converge on the same insight: well-curated KB models can generalize across populations and institutions.

Regarding the volume comparison, only the PTVs and left breasts’ volumes show statistically significant differences between the consortia test datasets. A larger spread of the MIKAPOCo left breast, if compared with MIKAPOCo PTVs, could be due to an inter-observer variation in the delineation of the left breast. This is not the case for the VPRSG test dataset. Nevertheless, in terms of PTV coverage, the VPSRG cohort shows similar values for PTVs of V95%, V105%, D1%, and D99% compared to most institutions of the MIKAPOCo training set dataset (Figure 2). A comparison of the obtained results with those described by Tudda et al. [28] reveals good agreement, particularly in terms of the mean values (averaged among the models) of predicted OARs mean doses and their standard deviations, expressing inter-institute variability on the dataset test cohort provided by the consortia. For instance, as shown in Table 1, the comparison of the ipsilateral lung mean dose between VPSRG (5.57 Gy) and MIKAPOCo (5.39 Gy), or heart mean dose (VPSRG: 0.59 Gy vs. MIKAPOCo: 0.39 Gy), demonstrates this alignment. However, the larger SD observed in the VPSRG patient test dataset highlights a higher inter-consortium variability in the prediction. This is evident not only for ipsilateral lung (VPSRG SD: 1.13 vs. MIKAPOCo SD: 0.55) but also for the heart (VPSRG SD: 0.31 Gy vs. MIKAPOCo SD: 0.17 Gy). The larger SD in VPSRG indicates increased variability across predictions, underscoring the importance of assessing and understanding the inter-consortium variability in the application of KB models. It is worth noting that despite volume variations, dose and OARs do not always correlate, and the prediction remains reliable and satisfactory even when volume differences occur between the test and training datasets. This can be attributed to the model’s reliance on principal component analysis, probably due to the consistent positioning of the PTVs in relation to the OARs.

This result is also confirmed when examining the ipsilateral lung PC1 to verify the transferability of KB prediction models. Consistent with the findings of Tudda et al. [28], the lowest model transferability in this study is associated with the Institute 6 model, with PC1 outside the 90th percentile in 39% of the patients. Similarly, during the VPSRG dataset test, PC1 values were outside the 90th percentile in 35% of the cases. Indeed, Institute 6 showed a model with no overlap between the PTV and the ipsilateral lung, leading to high PTV coverage and improved lung sparing [29]. While the Institute 6 model demonstrated agreement in both dataset tests in terms of reduced transferability, larger differences between the consortia for the ipsilateral lung PC1 transferability were observed for models from Institute 7 (20%), Institute 9 (15%), and Institute 10 (25%), as shown in Figure 5. Outlier inspection of the latter three institutes does not highlight that any relevant clinical-predicted discrepancies occurred, as in the case of Institute 6 (no overlap between PTV and ipsilateral lung), or similar, like possible larger PTV/left breast volumes or lower inclusion within training bounds. Model-specific patterns suggest that training set geometry probably contributes mostly to lower transferability. Increased evaluation of the dataset and eventually curating training sets for anatomical diversity may improve generalizability.

Given that the ipsilateral lung PC1 value outside the 90th percentile was observed in four out of ten models—still considered good transferability—this difference could be due to the difference in volumes.

KB prediction model performance was further quantified by grouping ipsilateral lung DVHs into four categories according to their consistency with the clinical plans. As depicted in the last figure in Appendix A, only 2% of cases exhibited failure (4 out of 200 DVH predictions). In 76% of cases, optimal prediction was achieved, indicating that the predicted ipsilateral lung DVH was within the expected range, aligning with the cDVH. This results underscores the excellent prediction performance of the models studied in this work. Moreover, in 15% of cases, the ipsilateral lung DVH prediction by MIKAPOCo’s KB models improved upon the cDVH, demonstrating potential for OAR sparing. Conversely, in only 7% of cases, the prediction was suboptimal compared to the cDVH, suggesting instances where the KB prediction model did not perform optimally. These interesting results translated into a lower, although relatively small, reduction (0.4–1.5 Gy) in the predicted mean lung dose compared to the clinical ones for 8 out of 10 models, prompting VPSRG to consider utilizing them for their planning practice. Outlier errors were uncommon but informative: larger residuals clustered in patients whose ipsilateral lung geometry lay outside the models’ training envelope (PC1 out of band) and in cases with atypical PTV–lung relationships relative to the originating cohort. Models trained without PTV–lung overlap (e.g., one institute, number 6) transferred worst to typical overlap anatomies, underscoring the role of geometry mismatch. Anatomical scale differences in the external cohort (larger PTV/left breast volumes) further widened the dispersion of Δ(cDVH-predicted). Despite these factors, the vast majority still met practical accuracy bands (88.7% of mean-dose Δ within ±2 Gy; 92.3% of V20 Gy Δ within ±5%), supporting overall portability. Pragmatic mitigations include training set enrichment at geometric extremes (overlap and size), and development of a unified, diversity-curated model to reduce out-of-envelope failures.

Moreover, possible implementation of the models in clinical practice deserves careful validation. For instance, many aspects could impact differences between predicted and calculated dose [19,42], such as machine configuration, dose algorithm, delivery technique, optimization template, and so on. We already studied and established this aspect through previous single-center investigations [43]. Although such investigation is beyond the scope of the present study, the deliverability of the predicted dose and its impact on the models’ transferability should be confirmed in a large-scale implementation and it will be the focus of a future study.

Even if this study is the first to assess national consortia KB models on RWB irradiation using TF, several limitations should guide future research. First, the VPSRG external cohort comprised 20 patients, adequate for feasibility/transferability assessment and validation but limiting precision at the patient level inference. Therefore, expanding it to other Australian regions and additional hospitals could provide more insights into geographical and methodological dependencies between consortia, enhancing the robustness and generalizability of the findings. Future studies could improve the results by using a unified MIKAPOCo model, collaboratively created by all MIKAPOCo institutions [43]. This standardized model would serve as a benchmark, offering better insight into KB model performance across different institutions and treatment scenarios. Second, the deliverability and end-to-end plan QA of automatically generated plans were not tested in this work here and warrant a prospective evaluation. Third, observed inter-consortium variability likely reflects differences in contouring practice, planning templates, and patient anatomy; while our PC1-based criteria are robust, they may not capture all clinically relevant nuances. Finally, predictions were evaluated against clinical DVHs rather than full re-optimizations, by design, to isolate model prediction performance. For this reason, plan deliverability and pretreatment QA of automatically generated plans were not assessed in this retrospective analysis due to the scope of the study. An ongoing prospective study using a pooled, generalized MIKAPOCo model [43] and a pre-specified on-machine feasibility pilot will address this question. Based on this achievement, a possible implementable framework could support some concrete and actionable strategies to mitigate inter-institute differences: standardized contouring, shared atlas, and one-page checklist, with quarterly 5-case inter-observer audits and predefined pass/fail thresholds; centralized QA and monitoring, validated on a 10-case common benchmark using harmonized tangential templates; require independent MU checks and EPID/array QA for pilot plans; track prediction residuals (Δ) with simple statistical process control to detect site drift; collaborative model refinement, to maintain a pooled base model with lightweight site adapters updated via a 10-case calibration set. As a last consideration, PTV/left breast differences could not be fully attributed because anthropometric covariates were not collected; prospective harmonization and covariate capture are planned in the next study.

The international inter-consortia experience highlights the potential of transferred learning in KB planning prediction models. By reusing models across domains, this approach accelerates robust model creation, benefiting institutions and consortia implementing automated planning in new or diverse clinical settings. This is particularly relevant for low- and middle-income countries with rising cancer cases and limited resources. Transferred learning improves model development, generalizability, and adaptability, while reducing the need for extensive data collection and addressing ethical challenges in data transfer, supporting efficient radiation therapy planning.

## 5. Conclusions

In this study, inter-consortia collaboration was proven to be not only feasible but also effective. For RBW-TF plans, comparable ipsilateral lung transferability was demonstrated in most of the large majority of the test cases and models, and good dose and statistical prediction were also demonstrated. The results of this work strongly support the use of national RWB-TF models for predicting RWB-TF outside of the national consortium where they were originally generated. These findings support international model sharing to promote plan quality consistency, reduce planning time, and enhance QA. Future work should prospectively evaluating deliverability on the treatment machines, expanding validation cohorts across additional regions and techniques, and investigating standardized, unified consortium models to further mitigate inter-institution variability.

## Figures and Tables

**Figure 1 cancers-17-03576-f001:**
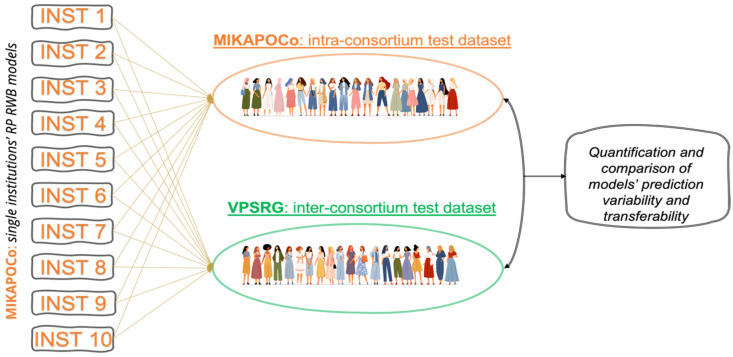
Ten KB models generated within the national consortium MIKAPOCo were evaluated and quantified in terms of variability and transferability on an intra-consortium dataset (MIKAPOCo) and an inter-consortium dataset (VPSRG).

**Figure 2 cancers-17-03576-f002:**
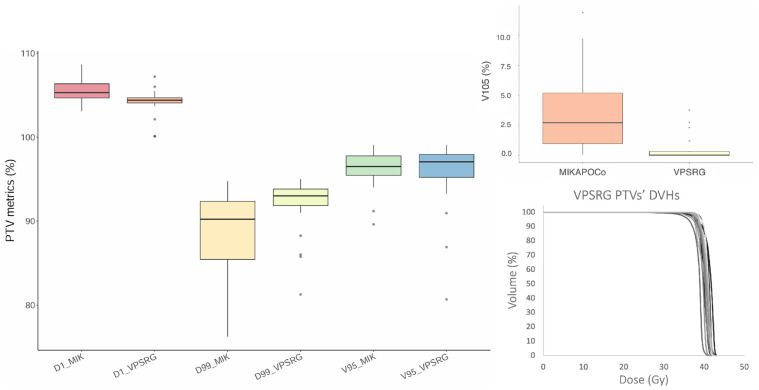
Box plot of V95%, V105% (**top right** panel), D1%, and D99% for the 20 MIKAPOCo’s test dataset (MIK) and the VPSRG’s test dataset PTVs. On the **right bottom**, the twenty PTV DVHs of the VPSRG’s test dataset are shown.

**Figure 3 cancers-17-03576-f003:**
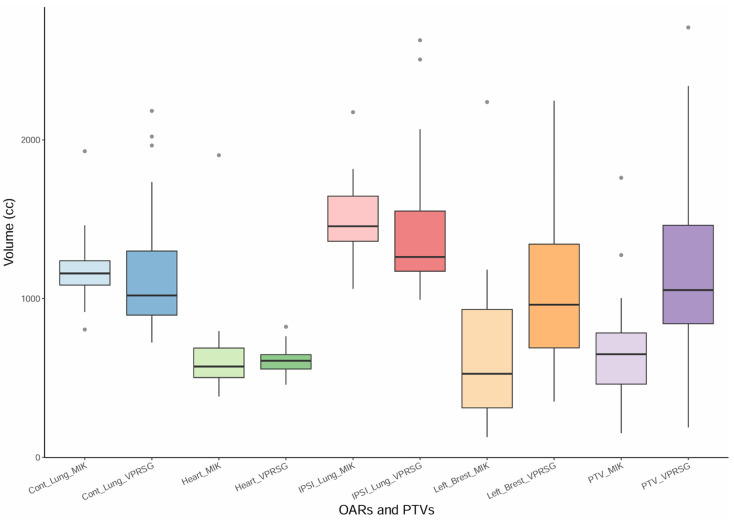
Box plot of the PTVs and OARs (contralateral lung, heart, ipsilateral lung, and left breast) volumes for the twenty VPSRG’s patients test datasets and the twenty MIKAPOCo’s patients test datasets, as used in Tudda et al. [26].

**Figure 4 cancers-17-03576-f004:**
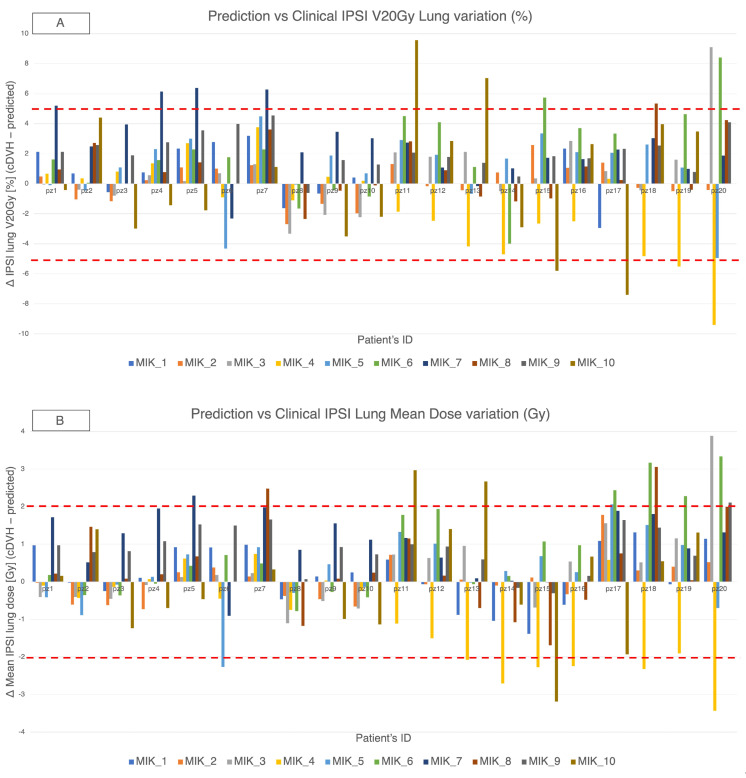
MIKAPOCO per-model differences (Δ = cDVH − predicted) for ipsilateral lung across 20 VPSRG patients. Top rows, (**A**) Δ V20 Gy (%); bottom rows, (**B**) Δ mean dose (Gy). Positive values indicate clinical > predicted. Dashed lines mark ±5% and ±2 Gy thresholds.

**Figure 5 cancers-17-03576-f005:**
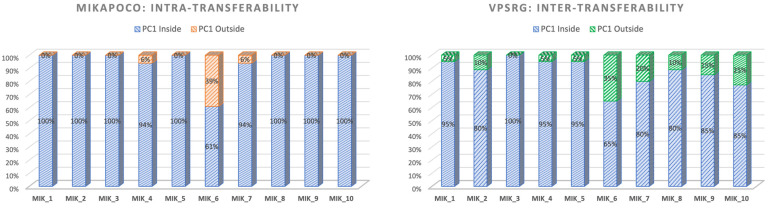
Values of ipsilateral lung PC1 for each consortium within the internal validation cohort for MIKAPOCo (**left**) and the external validation dataset for VPSRG (**right**). ‘‘Inside” refers to values less than or equal to the 90th percentile of the corresponding training set for all patients.

**Table 1 cancers-17-03576-t001:** Mean values (averaged among the models) of the predicted mean doses and their standard deviations, expressing the inter-consortium variability of the prediction, on the 20 test patients from VPSRG. These values are compared with those obtained in Tudda et al. [28], showing the intra-consortium variability.

OAR	Mean Dose_int_ ± SD_int_ (Gy)		Δ (VPSRG − MIKAPOCo)	*p* Value (Welch t, Two-Side)
	MIKAPOCo	VPSRG		
Heart	0.39 ± 0.17	0.59 ± 0.31	+0.2	0.095
Contralateral breast	0.26 ± 0.10	0.47 ± 0.31	+0.21	0.025
Ipsilateral lung	5.39 ± 0.55	5.57 ± 1.13	+0.18	0.658
Contralateral lung	0.12 ± 0.04	0.36 ± 1.13	+0.24	0.021

## Data Availability

Research data are stored in an institutional repository and will be shared upon request to the corresponding author.

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
