# Peer review of "An International Inter-Consortium Validation of Knowledge-Based Plan Prediction Modeling for Whole Breast Radiotherapy Treatment"

_cancers, 2025, doi:10.3390/cancers17213576_

Round 1
Reviewer 1 Report
Comments and Suggestions for Authors
This study validated KB prediction models developed in a national consortium using data from another multi-institutional consortium in a different country. Overall, this study is interesting and some what valuable.
Introduction: Please evaluate the incidence of breast cancer and the shortcomings of current treatment methods in order to highlight the value of this study.
In the Methods section, please refer to other researchers’ papers. The current methodology seems to be based on Tudda’s study.
Line 301–323: Please shorten these paragraphs. Focus the discussion on the findings of this study, rather than emphasizing research trends or background information.
Line 327–334: Please focus on the differences between Jain’s study and your study, or use his study to explain your own findings, instead of simply describing his work.
Please discuss the limitations or shortcomings of this study in the Discussion section.
Conclusion: Please explain the implications of this study for the field of breast cancer and discuss future research directions.
Some text should remain capitalized, such as Figure 1, Supplementary material 1.
Please keep the font consistent in the figures, for example, use Times New Roman for all graphical elements.
In Figure 2, the text is difficult to read; please make it bold, larger, or darker.
Please pay attention to punctuation usage. For example, there should be spaces before and after symbols such as “±” and “=”. In addition, other punctuation marks are also used inconsistently, please correct them.
Why does Figure 4 have no axes? Please add them.
Please update some of the references from 10 years ago, as these sources are too outdated.
Author Response
The point-by-point response to the reviewer number 1 has been uploaded. Please see the attachment.
Kind regards

Reviewer 2 Report
Comments and Suggestions for Authors
This manuscript addresses a clinically relevant and technologically advanced topic in radiation oncology with a rigorous methodology. It presents a well-structured and well-written study investigating the “geographical transferability” of knowledge-based (KB) radiotherapy planning models for whole breast cancer. The research question is novel, significant, and timely, as automated planning tools become more prevalent. The collaboration between two major international consortia, MIKAPOCO (Italy) and VPSRG (Australia), is a notable strength. The conclusion that KB models can be reliably shared between different countries is well-supported by the data and has substantial implications for standardizing and enhancing the quality of patient care globally.
Several significant enhancements can be implemented to elevate the quality of paper, as follows:
- The external validation study was conducted on a cohort of 20 patients. While this sample size may be sufficient for a feasibility study, a larger cohort would substantially enhance the strength of the conclusions and enhance the generalizability of the findings. The authors acknowledge this limitation, but it is a point that a reviewer would likely raise.
- The study indicates that four out of ten models exhibited reduced, albeit still acceptable, transferability (PC1 values fell within the desired range for approximately 65-85% of patients). To enhance the discussion, a deeper exploration of the reasons behind the inferior performance of these specific models (particularly those originating from Institutes 6, 7, and 10) would be beneficial. Could it be attributed to specific characteristics of their training data, such as unique patient anatomy or specialized planning techniques, that hindered their generalizability? A comprehensive analysis would yield valuable insights into the factors that contribute to the universal applicability of a KB model.
- Analysis Results: The analysis indicates that 88.7% of variations in the mean dose to the ipsilateral lung were below 2 Gy, and 92.3% of V20Gy variations were below 5%. These outcomes are favorable. However, the manuscript could benefit from a more comprehensive discussion of outlier cases where the predictions exhibited reduced accuracy. Are there discernible patient-specific anatomical characteristics that pose challenges to the models? Understanding the failure points is equally crucial as validating the successes.
- In the fourth section, the authors rightly point out that the practical feasibility of the automatically generated plans was not evaluated and should be the subject of a future study. Although this aspect exceeds the scope of the present work, reputable journals frequently prioritize completeness. Even a pilot study demonstrating that the plans for a limited number of representative cases are feasible on actual machines would substantially enhance the paper’s impact..
The minor correction is indicated as follows:
- In the abstract, “AIRC” should not be abbreviated on first mention unless it is a widely recognized term. Alternatively, the full name should be listed in the abbreviation section.
- In the abstract, the term “V20Gy” should be expanded upon its initial mention, followed by the abbreviation unless it is presumed that the majority of readers (e.g., >60%) are familiar with it.
- The introduction lacks clarity regarding the urgency of the study. The authors begin directly by discussing current breast cancer treatment. To enhance the impact, it would be beneficial to commence with a concise paragraph that succinctly highlights the severity of breast cancer, such as the mortality-to-prevalence ratio. Subsequently, emphasize the significance of radiotherapy in breast cancer treatment. Finally, proceed with the current introduction..
- The sentence “Currently, whole breast irradiation is a well-established therapeutic approach, with tangential fields (TF) being the predominant and widely utilized method, primarily using manual optimization” lacks impact. Consider strengthening it by emphasizing the limitations or consequences of manual planning, rather than simply stating it is labor-intensive.
- In the next paragraph, the authors jump directly to “knowledge-based optimization,” although the previous sentence mentions various automated approaches. It would be clearer to include a strong transitional sentence explaining why KB optimization was selected over other methods.
- Line 138 contains an unnecessary space please delete.
- It would be helpful to include a dedicated section explaining key terms such as MIKAPOCo, VPSRG, model prediction, variability, and transferability. Clarify what each represents, their roles in intra- vs. inter-consortium contexts, and from whose perspective these distinctions are made. For example, VPSRG could be considered intra if conducting its own study.
- In the methods section, it’s unclear how the 20 intra-consortium patients were selected. Was there a rationale or minimum sample size calculation to ensure representativeness?
- Similarly, for the inter-consortium VPSRG cohort, is a sample of 20 patients sufficient to represent the broader population?
- Figure 4 could be improved by breaking it into subfigures (e.g., 4A–4J), one per model. The current figure lacks clarity, and the results section does not provide detailed values, making interpretation difficult.
- The manuscript identifies statistically significant differences in PTV and left breast volumes between the MIKAPOCo and VPSRG cohorts (p = 0.002 and p = 0.02, respectively). However, the discussion offers only a brief mention of inter-observer variation as a possible cause. Given that other OARs show comparable volumes, a more detailed exploration of this phenomenon is warranted.
I recommend the authors expand their discussion to consider additional contributing factors, such as differences in patient demographics, contouring protocols, imaging modalities, or institutional practices. This would strengthen the interpretation of volume discrepancies and their potential impact on model transferability.
- The manuscript rightly identifies inter-institutional variability as a major limitation of knowledge-based planning, yet the discussion lacks depth in addressing how this challenge could be overcome. While strategies like unified model development and international comparisons are briefly mentioned, they remain conceptual. I suggest the authors strengthen the conclusion by offering more concrete and actionable strategies to mitigate inter-institute differences such as standardized contouring protocols, centralized QA frameworks, or collaborative model refinement which would enhance the robustness, generalizability, and clinical relevance of the proposed KBP approach.
Thank you
Author Response
The point-by-point response to the reviewer number 2 has been uploaded. Please see the attachment.
Kind regards

Reviewer 3 Report
Comments and Suggestions for Authors
- Line 317, 337 - To be statistical correct, whenever mentioned the word “significant” – please report the corresponding p-value to support such statement with the word of “significant”. If no p-value could be reported – suggest replacing with other words or phrases.
- Line 61 – “A cohort of 20 patients from the external consortium was used for testing” – I wonder what is the statistical power of a cohort 20 patients as a validation dataset? Strongly felt a validation sample size 20 (effect size) was too small to generate enough statistical power (power 80% and above) for any conclusions.
- Line 270: Table 1 – p-values of statistical tests for difference between MIKAPOCo vs VPSRGB need to be reported.
- Line 283: Figure 4 - p-values of statistical tests for difference between MIKAPOCo vs VPSRGB need to be reported.
- Line 393: Authors need to provide statistical results (for example, power analysis) to proof their statement “While the dataset size for VPSRG is adequate”. To me, a validation sample size 20 (effect size) was too small to generate enough statistical power (power 80% and above) for any conclusions.
- Line 393: Authors should clearly list more contents regarding what “several limitations” were.
- The contents of the paper are mostly just descriptive summaries. Lacking large enough sample size and multi-variable statistical tests results to proof the validity of study hypotheses.
Author Response
The point-by-point response to the reviewer number 3 has been uploaded. Please see the attachment.
Kind regards

Round 2
Reviewer 1 Report
Comments and Suggestions for Authors
Approved
Reviewer 3 Report
Comments and Suggestions for Authors
Author addressed most my questions and suggestions of the first review in this 2nd revision. No further comment/suggestion needed.